# Green and Eco-Friendly Treatment of Textile Wastewater by Using *Azadirachta indica* Leaf Extract Combined with a Silver Nitrate Solution

Muhammad Atif Irshad [1], Muhammad Ahmad Humayoun [1], Sami A. Al-Hussain [2], Rab Nawaz [1,*], Muhammad Arshad [3], Ali Irfan [4] and Magdi E. A. Zaki [2,*]

1  Department of Environmental Sciences, The University of Lahore, Lahore 54590, Pakistan
2  Department of Chemistry, College of Science, Imam Mohammad Ibn Saud Islamic University (IMSIU), Riyadh 13623, Saudi Arabia
3  Department of Agriculture and Food Technology, Karakoram International University, Gilgit 15100, Pakistan
4  Department of Chemistry, Government College University Faisalabad, Faisalabad 38000, Pakistan
*  Correspondence: rnuaf@yahoo.com (R.N.); mezaki@imamu.edu.sa (M.E.A.Z.)

**Abstract:** The present study was conducted to treat textile industrial wastewater through the combination of green and synthetic solutions. Two case studies were applied for the treatment of wastewater. In the first case, discharged industrial effluent was reacted with *Azadirachta Indica* leaf extract solution for a 4 to 72 h retention time. After the reaction, some pollutants were treated but most required higher retention time and concentration of *A. indica* extract, which could be a potential adsorbent for wastewater treatment. In the second case, the discharged industrial effluent was reacted with *A. indica* solution with silver nitrate $AgNO_3$ solution and was used as a treating agent for wastewater with a 4 to 72 h retention time. The second case was found to be better than the first case as it treated a greater number of pollutants. Moreover, treatment plant design feasibilities will be required for the application of findings of the present study on an industrial scale. This study can be useful to improve industrial estate's environmental conditions for reducing pollution by industrial wastewater. There is also a need to raise environmental awareness regarding wastewater's health effects in local communities.

**Keywords:** bio-adsorbent; textile wastewater; sustainable treatment; environmental management





## 1. Introduction

Water is essential for all forms of life and its sources include surface water bodies and groundwater. Surface water is found in rivers, lakes, and canals, whereas groundwater is found in wells and boreholes. Water is essential for drinking as well as for industrial and agricultural purposes. Water quality is being contaminated due to human activities, rapid population growth, unplanned urbanization, and industrialization globally and in Pakistan [1,2]. Water contamination is one of the biggest concerns being faced by most countries today. Water pollution has effects not only on the environment and human health, but also on society and the economy [3]. Many research studies has focused on a variety of nanomaterials, including molecular polymers (MIPs), organic and inorganic solutes of water-inducing bacteria, nanostructured catalytic membranes, nano sorbents, nano catalysts, bioactive nanoparticles, and biomimetic membranes, as well as the phytoremediation for the wastewater treatment [4–7].

There is rapid industrial development occurring worldwide that has negative consequences on all types of ecosystems and which cause environmental degradation. Tons of organic wastes are being discharged into aquatic environments as chemical spills and effluents every year [8]. Pollution has negative and severe impacts on climate, livestock, and humans. Rapid industrial growth has benefited humans in terms of employment and

provision of services, but it has also had unfavorable effects, including environmental deterioration and depletion of natural resources [8]. In addition to disturbing climatic conditions, pollution also has severely affected people and animals. Some manufacturers, including those in the food, cosmetic, pharmaceutical, and textile sectors, release toxic dye effluents into the environment after going through a number of chemical processes, affecting the ecology of habitats. In addition, chemical poisons harm the central nervous system and cause blood problems and skin irritation in both people and animals. Due to the rapid usage of pigments and dyes in the textile industry, large amounts can lead to a variety of health issues as well as eutrophication and acidification of water bodies. To address this issue, many wastewater treatment methods are available. Despite several water treatments and experimental results reported, there are still significant differences in opinions for better sustainable and environmentally friendly treatments [4,9].

The effluents from the dyeing industrial units have a higher level of BOD and COD, suspended particles, toxic substances, and a pigment that is visible to the human eye in extremely low quantities. Industrial waste can be used as an alternate method of treatment to adsorb colors from wastewater. To remove the chemicals from wastewater used to dye garments, physical, mechanical, and biological techniques are often utilized nowadays. These procedures also generate waste and are damaging to the environment [10].

Adsorption has been increasingly important in manufacturing and environmental conservation in recent years. Adsorbents are commonly used because of their environmental responsiveness, natural abundance, and low cost. Adsorption is a wastewater treatment method that removes a wide variety of contaminants from commercial wastewater. In biosorption, strong solid bio-adsorbents with a high porosity surface and the ability to remove heavy metal ions are used [11]. Due to their special properties, biosorbents are cost-effective adsorbents for removing heavy metals and other chemical parameters [12].

The *Azadirachta indica* tree belongs to the Meliaceae family and has been used to treat various human disorders and other hygienic practices since ancient times (especially its leaves and bark). The medicinal and germicidal properties of *A. indica* have been well-documented since ancient times. A fresh leaf of *A. indica* leaves produces a maximum of 59.4% moisture, 22.9% carbohydrates, 7.1% proteins, 6.2% fiber, 3.4% minerals, and 1% fats and other chemicals [13]. A variety of chemical compounds are present in *A. indica* extracts, such as nimbidin, nimbin, nimbolide, gedunin, azadirachtin, mahmoodin, cyclic trisulphide, and others are chemical components that are utilized as antipyretic, anti-inflammatory, antibacterial, anti-gastric ulcer, antiarthritic, spermicidal antifungal, antimalarial, hypoglycemic, immunomodulatory, diuretic, and anti-cancer agents [14,15]. This tree provides several environmental benefits that only a plant can give, such as protection of surrounding watersheds, enhanced stormwater management, moderation of the city heat island effect, reduction of noise pollution, and provision of social benefits. In recent years, certain trees, particularly *A. indica*, have been found for better air pollution mitigation along the roadside for better green belt development [16]. Previously, *A. indica* leaf powder was utilized as an adsorbent for the removal of various contaminants from their aqueous solutions for the study wastewater by investigators [17–19]. Furthermore, Ganguli (2002) characterized the structure and chemical composition of *A. indica* leaf powder as well as the adsorptive activities in wastewater treatment [20].

Because *A. indica* is widely dispersed throughout the world, including Pakistan, it was determined in the present study to use its leaf extracts as a bio-adsorbent for the treatment of textile wastewater. The main purpose of the study project was to examine the efficacy of *A. indica* mixed with a silver nitrate AgNO$_3$ solution from the textile industrial effluent. The efficacy of *A. indica* leaf extract in decreasing chemical effluents from the textile industry was evaluated by comparing the results to Punjab Environmental Quality Standards (PEQs). Details of various chemical parameters of wastewater are given in Table 1, with the standard values according to PEQS.

**Table 1.** Punjab Environmental Quality Standards (PEQS) for municipal and liquid industrial effluents.

| Sr. No | Parameter | PEQS Limits |
|:---:|:---:|:---:|
| 1 | COD (Chemical Oxygen demand) | 150 mg/L |
| 2 | pH | 6–9 |
| 3 | TDS (Total Dissolve Solids) | 3500 mg/L |
| 4 | TSS (Total Suspended Solids | 200 mg/L |
| 5 | BOD (Biochemical Oxygen Demand) | 80 g/L |

Source: https://epd.punjab.gov.pk/peqs (accessed on 14 September 2022).

## 2. Materials and Methods

### 2.1. Materials Used

The experiment was carried out in the springtime at 25 to 30 degrees and standard atmospheric pressure in the research lab of the Department of Environmental Sciences at The University of Lahore. The leaves of *A. indica* were harvested from the native area of Lahore and utilized to make the extract. A total of 0.05 M silver nitrate ($AgNO_3$), supplied by Sigma Aldrich, was employed, and all Pyrex equipment was used throughout the whole experiment after being cleaned with triple-distilled water.

### 2.2. Preparation and Application of A. indica Leaf Extract and Solution with $AgNO_3$

The leaves of *A. indica* were harvested in Lahore, Pakistan. Leaves were cleaned and cleansed using triple distilled water and absorbent paper. It was then mixed with ethanol and 100 mL of sterile water before being heated for 20 min at 70–80 °C. The extract was then filtered using the filter paper (Whatman's No.1 with pore size of 11 μm). The filtrate was obtained by regular sterilized filtration process and placed in a clear and dried conical flask [21]. Using the following cases, the obtained *A. indica* leaf extract was combined with $AgNO_3$ for the treatment of textile wastewater at a ratio of 200:10.

### 2.3. Treatment of Wastewater

Case 1: (Control): Pretreatment analysis of wastewater with 4 h, 8 h, 16 h, 24 h, 48 h, and 72 h Retention time.

Case 2: Addition of *A. indica* solution into a wastewater (200:10 *v*/*v*) ratio, i.e., 200 mL wastewater and 10 mL addition of *A. indica* extract.

Case 3: Addition of *A. indica* extract combined with silver nitrate ($AgNO_3$) solution, and afterwards, the solution was added to the pre-treated wastewater with a comparative ratio (200:10).

### 2.4. Analysis of Parameter and Post-Treated Wastewater

Wastewater was analyzed for different parameters including pH, COD, BOD, TDS, and TSS following the procedures as described in the 10th edition of Standard Methods for the Examination of Water and Wastewater [22]. The following equation(s) were used for the determination of various parameters of wastewater.

$$COD = \frac{(B - A) \times 8000}{\text{Volume of Sample}} \tag{1}$$

where, "A" is volume (mL) of FAS used in blank, "B" is ml of FAS used in the sample, and "N" is normality of FAS used.

$$TDS = \frac{(B - A) \times 10^6}{\text{Volume of Sample}} \tag{2}$$

where "A" is weight (gm) of empty dish and "B" is weight (gm) of beaker/dish and residue.

$$TSS = \frac{(B - A) \times 8000}{\text{Volume of Sample}} \tag{3}$$

where "A" is weight (gms) of GFC filter paper and "B" is weight (gm) of GFC filter paper and residue.

*2.5. Statistical Analysis*

All the data were analyzed for Analysis of Variance (ANOVA) using Statistical Software (SPSS, version 23.0 for windows; IBM Corporation, Armonk, NY, USA). Means values are presented in the graphs with standard deviation. Comparison was also made in graphs using the Punjab Environmental Quality Standards (PEQS).

*2.6. Reference Methods Used for Monitoring and Analysis*

Standard or Reference methods prescribed in the following published literature were used for sampling, onsite monitoring, mathematical calculations, and laboratory analysis regarding various environmental parameters. Furthermore, for comparison purposes, Punjab Environmental Quality Standards (PEQS) are given in Table 1.

## 3. Results

*3.1. Case 1: Pretreatment Analysis of Wastewater*

Pretreatment analysis of wastewater was done by measuring the average concentration of COD, pH, TDS, TSS, and BOD using the above-mentioned techniques. Comparison with PEQS values were made, all the results were beyond the permissible limit. The results of these parameters are 503 mg/L, 9.5 mg/L, 1765 mg/L, 365 mg/L, and 201.2 mg/L, respectively. All the results of these parameters are explained in Figure 1a–e.

### 3.1.1. Level of Pretreatment pH

The pH levels of the untreated effluent on day 1, day 2, and day 3 and the composite samples were 13, 12, 10, and 9.5, respectively (Figure 1a). Many of these values are out of range when compared with the PEQS. pH is one of the significant biotic factors serving as a pollution index. A textile effluent's pH value highly depends on the type of dye and material used for dying purposes, such as acidic, basic and reactive dyes and cotton, synthetic, etc., respectively [23]. For instance, elevated pH denatures the acid dyes and protein fibers. With respect to the textile industry, the highest COD values are mainly as a result of high coloring and various other textile operations.

### 3.1.2. Level of Pretreatment Chemical Oxygen Demand

Chemical oxygen demand levels are 460 mg/L, 485 mg/L, 510 mg/L, and 503 mg/L, respectively, for untreated effluents on day 1, day 2, day 3, and in the composite samples of all the days (Figure 1b). Values were compared with corresponding PEQS and it has been found that all these results are not within range. COD is strong in the textile industry because of elevated dyes. The textile industry generates significant volumes of wastewater, which contains high quantities of chemicals that are used during manufacturing processes [23]. These results also indicate the presence of organic matter in effluent waste, as shown in Figure 1b.

### 3.1.3. Level of Pretreatment TDS

TDS on day 1, day 2, and day 3, and in the composite samples were found to be 1700, 1750, 1820, and 1765, respectively (Figure 1c). Compared with PEQS, all these results were found to be higher than the permissible limit. Due to the presence of high dyes and other textile methods, the COD in the textile industry seems to be very elevated. At various stages during fabric processing, textile industries illustrate elevated TDS values as compared to other industries as a result of bleaching, dyeing agents, and fixing. The textile industry generates significant volumes of wastewater, which contain high quantities of chemicals added during processing processes [24]. TDS is considered a major factor in textile wastewater and as a result of glauber and common salt, TDS levels mainly increase in their effluent. Through direct textile wastewater discharge, TDS in surface and groundwater

may also increase. A high TDS value may have an impact on osmotic balance, alteration in taste, and aquatic life dehydration in water [25]. These findings showed the presence of organic matter in effluent, as presented in Figure 1c.

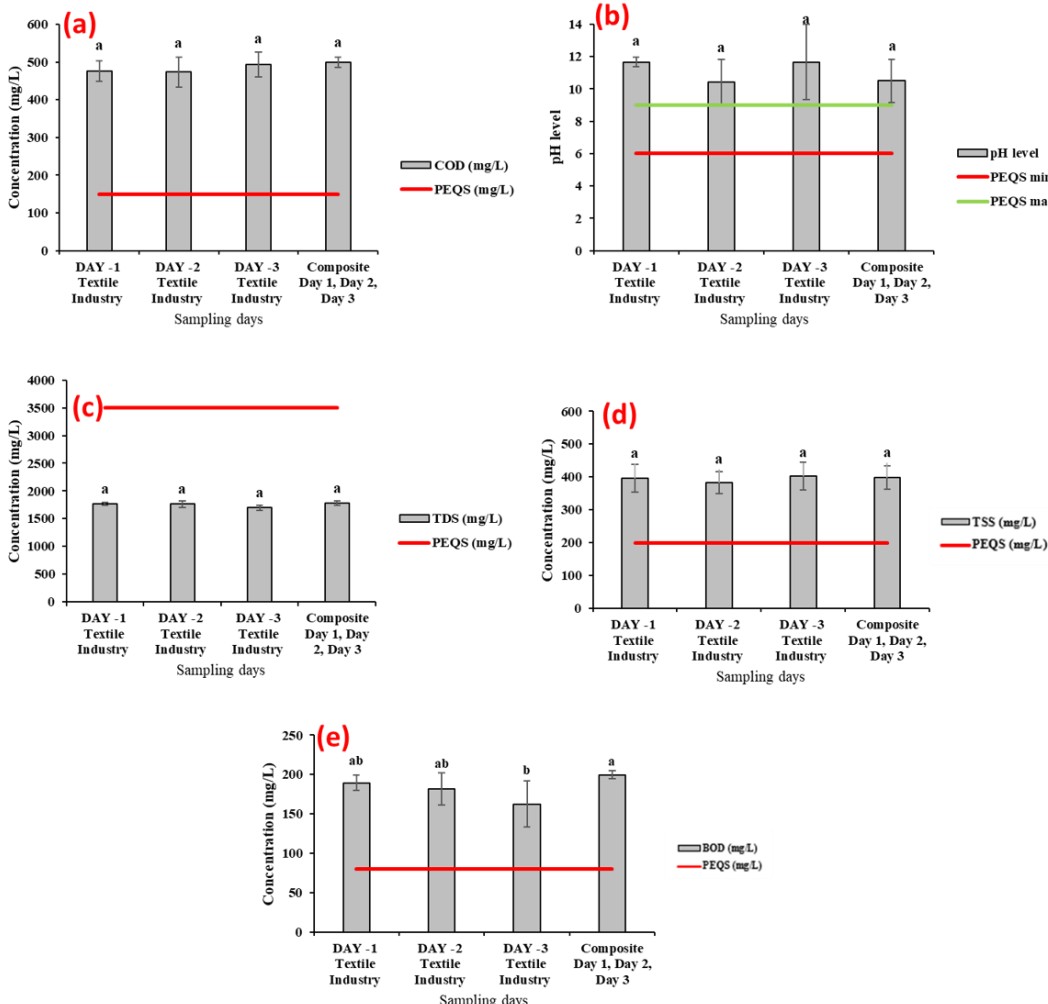

**Figure 1.** In (**a**), COD level of wastewater samples before the treatment, (**b**) pH level of pretreated samples of wastewater; all results also exceed the standards, (**c**) total dissolved solids (TDS) level of pretreated samples, (**d**) total suspended solids (TSS) level of the samples before treatment, which was also found to be higher than the permissible level recommended by PEQS, and (**e**) BOD level of all the samples of textile wastewater of the pretreated samples. All samples have higher values then the permissible limit. Different letters (e.g. a, b, etc.) shows significant different between treatments. Same letters (e.g. a, a) indicate no significant difference between treatments.

### 3.1.4. Level of Pretreatment TSS

The effluent's non-treated TSS level of day 1, day 2, day 3, and in the composite samples are 355, 395,435, 365, respectively (Figure 1d). Suspended solids appear to be significant pollutants in wastewater. Owing to excessive dyes and other textile methods, the COD in the textile industry is quite high [26]. It also demonstrate that textile industry effluent is high in suspended solids (SS), chemical oxygen demand, and temperature. Thus, it is significant to monitor parameters values with corresponding standards during treatment processes before discharging the effluent into a receiving water body.

### 3.1.5. Level of Pretreatment BOD

Normally, textile industries consume organic substances such as raw materials, and elevated dissolved organic matter utilizes high oxygen levels and increases the level of

BOD. By this high BOD value, a state of hypoxia with consequentially severe impacts can affect aquatic life. In the present study, BOD values of 184, 194, 204, and 201 mg/L are found for untreated day 1, day 2, day 3 effluent and the in the composite sample of all the days, respectively. It is clear that all values are far beyond PEQS. Because of various operations and processes being followed in the textile industry and with applications of dyes, the COD of wastewater of this sector is high. Figure 1e demonstrates the BOD values of all the samples.

### 3.2. Case 2: Reduction of Pollutant Parameters by Addition of A. indica Extract in Contaminated Sample

Experiment was performed in the lab with different retention times to reduce the level of pollutant at 4 h, 8 h, 16 h, 24 h, 48 h, and 72 h retention times for the treatment of textile wastewater. Firstly, a 200 mL sample was taken in the glass beaker and 10 mL of *A. indica* solution was added to the contaminated sample. Afterwards, the sample was stirred in the stirring chamber. As per Figure 2a–e, results show that all the pollutants were treated by the *A. indica* solution and reduced concentration of the parameters were found in comparison with the control (pretreatment samples) but could not lower the values below PEQS. The *A. indica* solution required more retention time for further treatment to obtain better results. There are many research studies on the efficacy of locally available natural coagulants such as neem, moringa, and various other plant extracts for lowering levels of wastewater pollutants to improve their quality, but the use of *A. indica* as a bio-adsorbent is also an effective and economical adsorption method to treat textile wastewater to reduce environmental damages [26–29].

### 3.2.1. Treatment of COD by Addition of *A. indica* Solution

As per the above table and figure, results showed that *A. indica* solution served as a treating agent for the reduction of COD, by the addition of 10 mL *A. indica* solution in the contained sample with different retention times. In the pretreated sample, the COD result was 503 mg/L, but by the addition of *A. indica* solution in the composite contaminated sample, COD was reduced in the 4 h retention time and in the 8 h, 16 h, 24 h, and 48 h retention time up to 72 h; the results were 485, 455, 365, 275, 195, and 165 mg/L, respectively. These results indicated that COD was reduced by the addition of *A. indica* solution but cannot achieve the permissible level of Punjab Environmental Quality Standards (PEQ's) (Figure 2a). As per above the condition, a greater retention time is required for the reduction of COD for the 10 mL addition in a 200 mL contaminated sample. After 72 h, COD was reduced, but greater time and cost are required to achieve the desired results. Natural coagulants including *A. indica* can be extracted from plants and are basically organic-based and used to reduce COD, turbidity, color, and organic matters [26,27].

### 3.2.2. Treatment of pH by Addition of *A. indica* Solution

The findings showed that *A. indica* solution acts as a treatment agent for pH reduction, as shown in the above table and figure. The pH results for the pretreated sample were 9.9 but the addition of *A. indica* solution to the composite contaminated sample decreased the pH the 4 h retention time and in the 8 h, 16 h, 24 h, and 48 h retention time to 72 h. These findings suggested that pH was decreased by the addition of *A. indica* solution and the permissible level of PEQS (Figure 2b). As reported in studies [30], the maximal adsorption potential of *A. indica* is affected by a variety of operating parameters such as pH, adsorption particle size, and activation. Hence, for pH, no further retention time is needed for pH level reduction. The pH level was decreased at 4 h retention time. *A. indica* is the strongest indicator for reducing the amount of pH for wastewater equalization if wastewater is of a basic type. A similar finding was found in another study [29,30], which evaluated the efficacy of locally accessible natural *A. indica* leaves powder to control characteristics of textile wastewater for improvement of its quality and the effect of various experimental conditions such as early pH of the wastewater samples.

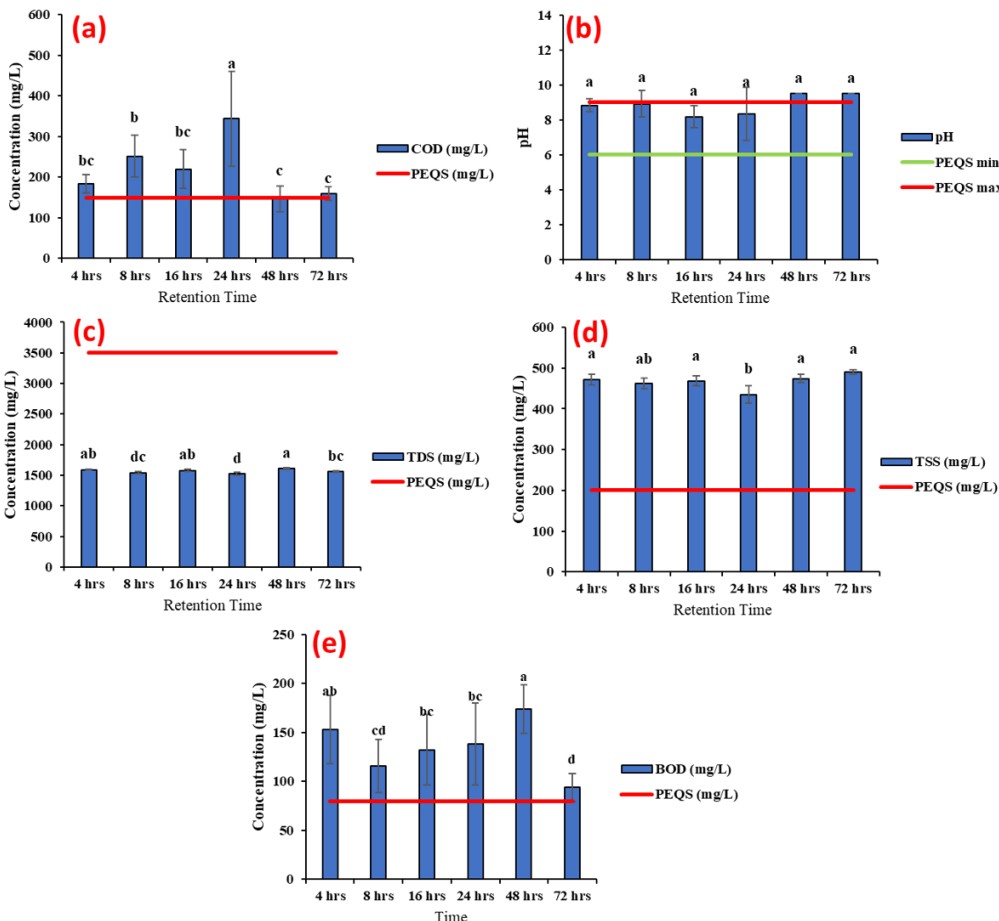

**Figure 2.** (**a**) the expression of the pH level of all the samples treated with *A. indica* extract, found to be effective to control pH towards the permissible limit, (**b**) COD treatment of waste water samples with the *A. indica* leaf extract. It indicates that COD of the samples reduced with the increase in retention time, (**c**) TDS level of the samples treated with the *A. indica* extract, found to be effective to control the total dissolve solids from all the wastewater samples, (**d**) TSS level of samples with the *A. indica* leaf extract, indicating a minimal effect on controlling the total suspended solids through the wastewater samples, and (**e**) BOD level of all the samples taken from the three consecutive days and treated with *A. indica* extract. Figure explains that *A. indica* extract has a minimal effect on BOD levels of the samples. Different letters (e.g. a, b, c, etc.) shows significant different between treatments. Same letters (e.g. a, a) indicate no significant difference between treatments.

### 3.2.3. Treatment of TDS by Addition of *A. indica* Solution

As shown in Figure 2c, the findings showed that *A. indica* solution acts as a treatment agent for the reduction of TDS. A total of 10 mL of *A. indica* solution was added to the required sample with varying retention times. The result of the pre-treated sample TDS was 1765 mg/L but with the addition of *A. indica* solution to the composite contaminated sample, TDS decreased at the 4 h retention period and in the 8 h, 16 h, 24 h, and 48 h retention time up to 72 h. Results were 1625, 1605, 1595, 1575, 1562, 1550 mg/L, respectively, suggesting that TDS had decreased with the addition of solution. As per the above condition, a greater retention time is required in order to reduce the TDS by 10 mL into the addition of 200 mL of the contaminated sample. After 72 h, the TDS was reduced, but more time and homogenization will be needed to achieve the desired results.

### 3.2.4. Treatment of TSS by Addition of *A. indica* Solution

The results show that *A. indica* solution serves as a treatment agent for reducing TSS, as seen in the 2d. A total of 10 mL of *A. indica* solution was added in a separate retention time

sample. Pretreated TSS was 365 mg/L, but by the addition of *A. indica* to the composite-contaminated sample TDS, retention time was high for four hours at 8 h, 16 h, 24 h, 48 h, and up to 72 h. The findings showed that the TSS increased to 465, 472, 485, 485, 489, 495 mg/L, respectively, by the addition of A. indica, without achieving a permitted Punjab Environmental Quality Standards (PEQS) level. As mentioned above, a further time for the retention of the TDS is appropriate in a 200 mL contaminated sample with an addition of 10 mL. TSS is raised after 72 h.

### 3.2.5. Treatment of BOD by Addition of *A. indica* Solution

*A. indica* solution acts as a BOD reduction treatment agent, as reported in the findings and shown in the above figure and table. A total of 10 mL of *A. indica* solution was added with varying retention times to the sample. The pretreated TDS sample resulted in 201.2 mg/L, but the addition of *A. indica* solution to the composite polluted the TDS sample and decreased the retention period by 4 h to 8 hrs, 16 h, 24 h, 48 h and 72 h. The findings were 194, 182, 146, 110, 89 and 82 mg/L, respectively (Figure 2e). These results showed that TDS has decreased by applying *A. indica* solution, but generally the permissible amount of Punjab Environmental Quality Standards (PEQS) had not been achieved. Greater retention time is required for the reduction of TDS at a 10 mL addition in a 200 mL polluted sample as per the above situation. TDS can be decreased after 72 h but more time and homogenization are required to achieve the desired outcome.

### 3.3. Case 3: Reduction of Pollutant Parameters by the Addition of A. indica Extract and Silver Nitrate 0.005 M Solution

An experiment was performed in the lab with different retention times to decrease the level of pollutants at different retention times such as 4 h, 8 h, 16 h, 24 h, 48 h and 72 h, respectively. Firstly, 200 mL of sample combined with 10 mL *A. indica* solution were mixed in the presence of 1 mL of $AgNO_3$ (0.05M). The mixture of all these constituents were stirred in the stirring chamber. As per Figure 3a–e, results showed that all the pollutants were well treated by the *A. indica* and silver nitrate solution. It has been observed that this is an easy and cost-effective method to treat and achieve the Punjab Environmental Quality Standards (PEQS). From textile dye wastewater, the degradation was also studied [24] at various initial pH of solutions with findings between the 5–8 range. As reported, it was found that *A. indica* induced silver nanoparticles through green synthesis as a capping and reducing agent. Silver nitrate, *A. indica* solution, and an NaOH base solution are suitable solutions for wastewater treatment at a 24 h retention time, but the level of pH did not decrease to the PEQS level.

### 3.3.1. Post-Treatment Results by Addition of *A. indica* Solution and 0.05 M Silver Nitrate in the Contaminated Sample (Wastewater)

As seen in the table and figure above, results illustrate that an *A. indica* solution and silver nitrate were used for the reduction of COD by the addition of 10 mL of *A. indica* solution in the contained sample with 1 mL silver nitrate at different retention times. In the pre-treated sample, the COD results were 503 mg/L, but by the addition of *A. indica* in 1 mL of silver nitrate ($AgNO_3$) in the composite contaminated sample COD was reduced at a 4 h retention time and at 8 h, 16 h, 24 h, and 48 h retention time up to 72 h. Results were 183, 134, 124, 92, 41, and 27 mg/L, respectively. These results indicated that COD was reduced by the addition of *A. indica* solution but at 8 h, the retention time results achieved the permissible level of PEQS. As per above condition, a cost-effective treatment can be achieved for long term results. Treatment models will need to be designed for further research and treatments. However, in water treatment technology, using various techniques, adsorption is the most protuberant, particularly in the removal of dyes. The high demand for cost effective and sustainable treatment practices has created an upsurge in bio-adsorbents' significance [31,32].

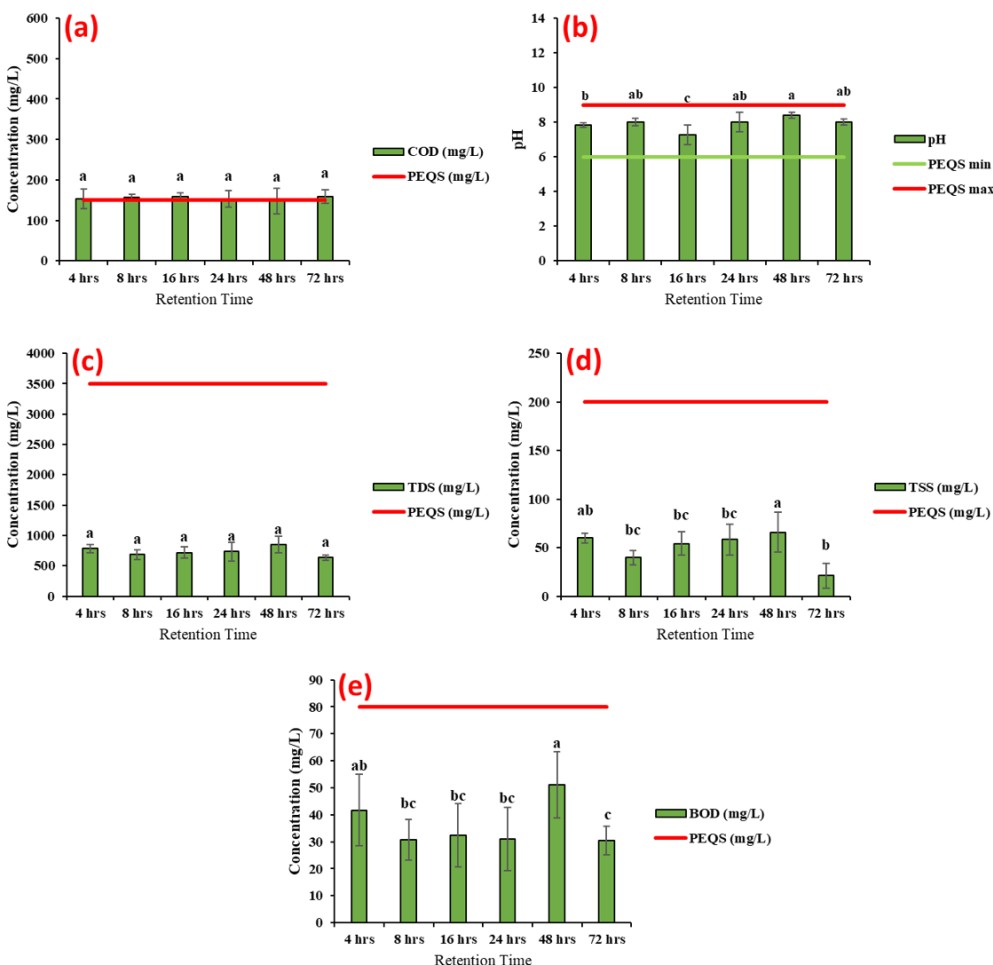

**Figure 3.** (**a**) pH treatment of all the samples through the combination of *A. indica* extract and AgNO₃, (**b**) treatment of COD with AgNO₃ combined with *A. indica* extract, (**c**) total dissolved solid treatment of the samples with the combination of *A. indica* leaf extract with AgNO₃, (**d**) total suspended solid treatment of wastewater by the combination of AgNO₃ and *A. indica* leaf extract, and (**e**) the treatment of BOD of all the textile wastewater samples with the combination of *A. indica* leaf extract and AgNO₃ solution. From all the figures, it is concluded that the combination of both AgNO₃ and *A. indica* extract have very strong effect on the physiochemical parameters of textile wastewater. Different letters (e.g. a, b, c, etc.) shows significant different between treatments. Same letters (e.g. a, a) indicate no significant difference between treatments.

### 3.3.2. Post Treatment of pH by Addition of *A. indica* and 0.05 M Silver Nitrate in Contaminated Sample (Wastewater)

The findings shown in the table above indicated that the AgNO₃ solution used for pH reduction by *A. indica* in a 10 mL of solution with 1 mL of AgNO₃ in the contaminated sample at varying retention times was effective. The pretreatment pH results were 8.5, 8.5, 8.3, 8.1, 8.1, and 7.8. However, the pH was increased at a 4 h retention time by incorporating this solution into the composite contaminated sample. Results were optimum at all retention times. The permissible level of PEQS can therefore be achieved, as the experiment demonstrated the change in pH with different retention times.

The findings revealed, as seen in the Figure 3a, that the *A. indica* solution acts as a TDS reduction treatment agent. TDS was 1765 mg/L in the pretreated sample result. However, by applying the *A. indica* solution and silver nitrate to the composite infected TDS sample, the retention period was 4 h instead of 8 h, 16 h, 24 h, and 48 h, up to the 72 h retention time. Results for all the samples were observed as 985, 845, 712, 685, 625, and 602 mg/L, respectively. These results suggested that TDS decreased with the addition of *A. indica*

solution and achieved the permissible level as per PEQS. As per the above situation, greater retention time is needed to reduce the TDS by 10 mL by the addition of 200 mL of polluted sample. After 72 h, the TDS will be reduced.

3.3.3. Post Treatment of BOD by Addition of *A. indica* and 0.05 M Silver Nitrate in Contaminated Sample (Wastewater)

According to Figure 3e, the results demonstrated that the *A. indica* solution combined with $AgNO_3$ acts as a BOD treatment agent. As per the pre-treated samples findings, BOD values were in the range of 201.2 mg/L at 4 h, 8 h,16 h, 24 h, 48 h, and 72 h retention times. When *A. indica* and $AgNO_3$ solution was applied, BOD was found to be 65.2, 45.8, 42.4, 36.4, 15.2 and 10.4 mg/L, respectively, at the above given retention times. The results demonstrated that BOD is reduced to an acceptable PEQS level by the addition of *A. indica* solution and silver nitrate. Accordingly, 4 h of retention time for BOD reduction in 200 mL of a contaminated sample is not desirable as a proper treatment as it has very low efficiency. Similarly, plant material has been used for the treatment of wastewater in other studies, such as activated carbon from avocado peels for the reduction of biological oxygen demand (BOD) and chemical oxygen demand (COD) [33], and MgO nanopowders (synthesised using Carica papaya leaf extract) for the reduction of COD and BOD in raw tannery wastewater [34].

**4. Conclusions**

Treatment and reuse of wastewater helps to protect and maintain the water balance but also raises concerns. Many studies have been undertaken on the protection of environmental health and flora and fauna in the ecosystem. Green and chemical solution-oriented research with special physical and chemical properties has tremendous potential for pollutant removal. At a 4 to 72 h retention time of the discharged industrial effluent with *A. indica* solution, some pollutants were treated but most pollutants required a greater retention time and concentration of *A. indica* extract for the treatment of wastewater. However, for the 4 h retention time of the discharged industrial effluent with *A. indica* solution along with silver nitrate ($AgNO_3$) solution, most of the pollutants were treated at a high pH of 13. It was found that more retention time and a better solution were required for the complete treatment of pollutants. However, the second case, i.e., *A. indica* extract combined with $AgNO_3$ was found to be much better than the first one. More pollutants were treated and further research should be done for the treatment of wastewater on a large industrial scale.

For the best outcomes in wastewater purification, treatment design is necessary for industrial-scale use. More research is needed to address the issues of green and chemical wastewater management systems. Primary and secondary applications are necessary for the use of synthetic and green solutions in wastewater treatment. More research on the economics and scalability of wastewater treatments using green and synthetic chemicals are needed. For the application of industrial ecology, an engineering management system should be established and implemented on the industrial scale.

**Author Contributions:** Conceptualization, M.A.I., M.A.H. and R.N.; methodology, M.A.I. and M.A.H.; software, M.A.I. and M.A.H.; validation, A.I., M.A. and M.A.H.; formal analysis, M.A.I., M.E.A.Z. and M.A.H.; investigation, M.A.I., M.A.H. and A.I. resources, M.A.I., S.A.A.-H. and M.A.H.; data curation, M.A.I., S.A.A.-H., M.A.H. and M.A.; writing—original draft preparation, M.A.I. and M.A.H.; writing—review and editing, A.I., R.N. and M.E.A.Z.; visualization, M.A.I. and M.A.H.; supervision, M.A.I. and R.N.; project administration, M.A.I., M.E.A.Z. and R.N.; funding acquisition, A.I. and S.A.A.-H. All authors have read and agreed to the published version of the manuscript.

**Funding:** The authors thank the Deanship of Scientific Research at Imam Mohammad Ibn Saud Islamic University for funding this work through Research Group no. RG 21-09-76.

**Institutional Review Board Statement:** Not applicable.

**Informed Consent Statement:** Not applicable.

**Data Availability Statement:** All the data have been provided in the manuscript.

**Acknowledgments:** The authors are very thankful to the Department of Environmental Sciences, The University of Lahore, for the completion of this research project.

**Conflicts of Interest:** The authors declare that there is no conflict of interest.

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
