# Peer review of "Green and Eco-Friendly Treatment of Textile Wastewater by Using Azadirachta indica Leaf Extract Combined with a Silver Nitrate Solution"

_sustainability, doi:10.3390/su15010081_

Round 1

Reviewer 1 Report

The methodology is not clear. It is not indicated where the reaction was performed, nor the stirring and temperature.

The A. indica extract is not analyzed, there is no information about the concentration of active compounds in the extract.

There is no indication of the chemical composition of the A. indica extract and how it works in the treatment of the wastewater. It is a chemical reaction, adsorption, coagulant, oxidant?

It is stated that the treatment is eco-friendly, but the use of A. indica alone is not appropriate to treat the wastewater. It was required to combine A. indica with silver nitrate and sodium hydroxide, which are expensive and toxic chemicals. There is no information of what is the effect of each of these compounds on the reduction of COD, TSS…. It seems that all the “pollutant reduction” was due to the addition of silver nitrate and sodium hydroxide.

I would recommend an extensive revision of the methodology and discussion of the results, otherwise, I would not accept the present manuscript for publication.

Author Response

Dear Reviewer! Thanks for useful comments for the improvement of manuscript. Manuscript has been revised as per comments/ suggestions of reviewer. Overall quality of manuscript has been improved. Language has also been improved. All the changes are yellow highlighted in the manuscript. Responses against all comments are given in the attached file.

Thanks and regards

Authors

Reviewer 2 Report

First of all, the paper does not have the appropriate structure which is essential in order to consider it for the review process.   Secondly, the references list is only 26 , is irrelevant and very superficial.   Third, the Conclusion section is represented by 3 brief paragraphs which is too scarce.   Fourth, the paper presents 3 chemistry case studies solely. the paper doesn't have a suitable academic approach.   No literature review, referring to previous similar research.   No research purpose, question, or objectives.   The paper doesn't fit at the moment.   The references list is missing a lot.

The paper must be reviewed in terms of structure, and research approach.

Author Response

(The authors gave the same response as above.)

Round 2

Reviewer 1 Report

The authors did improve the manuscript from the original version, but there were still important points without consideration. Although I have doubts about the quality of the present paper, I leave it for the Editor to decide.

Page 2

“These methods are, however, excessively expensive”

Which methods? Please include also a reference for this statement.

Page 3

filter paper Whatman’s No.1.” What is the pore size of the filter?

Page 4

The table should get a number and a caption. It is not clear what is presented in this table. Is it the limits of each parameter for wastewater disposal?

Page 7

It is stated that A. indica is a bio-adsorbent. But it is also stated the COD was reduced because A. indica is a coagulant. In any case, there is no evidence to prove that any of these mechanisms is happening. If there is sedimentation of the adsorbent or coagulation of the pollutants, the treatment should include a solid-liquid separation step, which is not discussed in the manuscript.

Author Response

Reviewer-1: Round 2

Dear Reviewer, Manuscript has been revised according to the following minor comments. All changes for second revision are highlighted (green) in the manuscript.

Response to Comments and Suggestions for Authors

Sr. No.

Comment/ Suggestion

Response/ Revision

1

Page 2

“These methods are, however, excessively expensive”

Which methods? Please include also a reference for this statement.

Dear reviewer, thank you for pointing out the critical issue in the text. This is a worthless sentence that was inserted mistakenly. It has now been deleted from the text.

2

Page 3

 “Filter paper Whatman’s No.1.” What is the pore size of the filter?

The pore size of the Whatman’s filter paper has been written now at relevant place in the manuscript.

3

 Page 4

The table should get a number and a caption. It is not clear what is presented in this table. Is it the limits of each parameter for wastewater disposal?

Thank you for bringing up the huge table inadequacy. The manuscript now includes a revised table with source information.

4

Page 7

It is stated that A. indica is a bio-adsorbent. But it is also stated the COD was reduced because A. indica is a coagulant. In any case, there is no evidence to prove that any of these mechanisms is happening. If there is sedimentation of the adsorbent or coagulation of the pollutants, the treatment should include a solid-liquid separation step, which is not discussed in the manuscript.

There is a penalty of research that has been published on the efficacy of locally available natural coagulants such as neem, moringa, and various other plant extracts in lowering wastewater characteristics to improve its quality. Kindly see references 26-27. These sentences are added in page 7.

Dear worthy reviewer, Authors provided response to all of your comments, I hope now this manuscript will fulfill criteria of acceptance of well esteemed journal  sustainability.

Thanks and regards

Authors

Reviewer 2 Report

Dear Authors,

your improvement effort are well-done.

Improvements are ok.

References have been improved but still seem to be less than the average.

Conclusions have been improved but still seem to be more concise than the average.

Author Response

Reviewer-2: Round 2

Dear Reviewer, Manuscript has been revised according to the following minor comments. All changes for second revision are highlighted (green) in the manuscript.

Response to Comments and Suggestions

Sr. No.

Comment/ Suggestion

Response/ Revision

1

References have been improved but still seem to be less than the average.

Conclusions have been improved but still seem to be more concise than the average.

Dear reviewer, as much as possible, the best references related to this study have been cited in the text.Further, more relevant studies have been cited in the results and discussion (page 10) and their full references (33-34) are also added in reference list.

Conclusion section has been updated with more possible results and recommendations.

Dear worthy reviewer, Authors provided response to all of your comments, I hope now this manuscript will fulfill criteria of acceptance of well esteemed journal sustainability.

Thanks and regards

Authors
